# Primary exposure to SARS-CoV-2 variants elicits convergent epitope specificities, immunoglobulin V gene usage and public B cell clones

Noemia S. Lima [1,7], Maryam Musayev[1,7], Timothy S. Johnston [1,7], Danielle A. Wagner [1,7], Amy R. Henry[1], Lingshu Wang[1], Eun Sung Yang [1], Yi Zhang[1], Kevina Birungi[1], Walker P. Black[1], Sijy O'Dell[1], Stephen D. Schmidt[1], Damee Moon [1], Cynthia G. Lorang[1], Bingchun Zhao[1], Man Chen[1], Kristin L. Boswell[1], Jesmine Roberts-Torres[1], Rachel L. Davis[1], Lowrey Peyton[1], Sandeep R. Narpala[1], Sarah O'Connell[1], Leonid Serebryannyy[1], Jennifer Wang[1], Alexander Schrager[1], Chloe Adrienna Talana [1], Geoffrey Shimberg[1], Kwanyee Leung[1], Wei Shi[1], Rawan Khashab[2], Asaf Biber [2,3], Tal Zilberman[2,3], Joshua Rhein [4], Sara Vetter[5], Afeefa Ahmed[4], Laura Novik[1], Alicia Widge[1], Ingelise Gordon[1], Mercy Guech[1], I-Ting Teng[1], Emily Phung[1], Tracy J. Ruckwardt [1], Amarendra Pegu [1], John Misasi [1], Nicole A. Doria-Rose[1], Martin Gaudinski[1], Richard A. Koup[1], Peter D. Kwong [1], Adrian B. McDermott [1], Sharon Amit[6], Timothy W. Schacker [4], Itzchak Levy[2,3], John R. Mascola[1], Nancy J. Sullivan[1], Chaim A. Schramm [1] ✉ & Daniel C. Douek [1] ✉

An important consequence of infection with a SARS-CoV-2 variant is protective humoral immunity against other variants. However, the basis for such cross-protection at the molecular level is incompletely understood. Here, we characterized the repertoire and epitope specificity of antibodies elicited by infection with the Beta, Gamma and WA1 ancestral variants and assessed their cross-reactivity to these and the more recent Delta and Omicron variants. We developed a method to obtain immunoglobulin sequences with concurrent rapid production and functional assessment of monoclonal antibodies from hundreds of single B cells sorted by flow cytometry. Infection with any variant elicited similar cross-binding antibody responses exhibiting a conserved hierarchy of epitope immunodominance. Furthermore, convergent V gene usage and similar public B cell clones were elicited regardless of infecting variant. These convergent responses despite antigenic variation may account for the continued efficacy of vaccines based on a single ancestral variant.

Over the course of the SARS-CoV-2 pandemic, selective immune pressure is proposed to have led to the accumulation of changes in residues targeted for antibody recognition and neutralization, most importantly in the receptor binding domain (RBD)[1,2]. While CD4 and CD8 T cell responses do not seem to be substantially impacted by variant substitutions[3], the neutralizing capacity and some Fc-mediated functions of antibodies induced by the ancestral SARS-CoV-2 variant (WA1) are significantly reduced against later

variants[4,5]. Despite this, first-generation vaccines based on the WA1 sequence continue to provide protection from severe disease and death[6], even against antigenically distant variants such as Delta (PANGO lineage B.1.617.2) and Omicron (B.1.1.529). The mechanism of this cross-protection is not fully understood at the molecular level, even though the humoral response to the ancestral virus has been well characterized[7–10]. Notably, the response to ancestral WA1 is highly consistent and includes polarization toward specific IG $V_H$ genes[11–13] and convergent V(D)J rearrangements ("public clones") found in multiple individuals[13–15].

In this work, we leverage high-resolution analysis of the immune responses to other, antigenically divergent, variants to explore the extent of conservation of these responses and to shed light on mechanisms of cross-protection. In a cohort of convalescent individuals infected with WA1, Beta (B.1.351), or Gamma (P.1), we use a novel method for rapid-throughput, cloning-free recombinant mAb synthesis and sequencing to investigate epitope targeting, $V_H$ gene usage, and B cell clonal repertoires against these variants, as well as Delta and Omicron. We show that the responses to primary infection with the three variants investigated here are convergent across multiple dimensions, which may provide a mechanistic basis for the observed cross-protection.

## Results

### Binding and neutralization titers

We collected serum or plasma and peripheral blood mononuclear cells (PBMC) from individuals infected with WA1, Beta, or Gamma variants 17–38 days after symptom onset (Supplementary Table 1) to compare antibody and B cell responses. All individuals had no known previous exposures to SARS-CoV-2. To focus on the total antigen-specific B cell repertoire, we selected samples from early convalescence, when frequencies of B and T cells are typically high, irrespective of neutralization titers.

We measured serum binding titers to variant spike (S) protein expressed on the surface of HEK293T cells (Fig. 1A) and to soluble stabilized variant S trimers (S-2P) and RBD using a Meso Scale Discovery electrochemiluminescence immunoassay (MSD-ECLIA) (Supplementary Fig. 1A). While MSD-ECLIA is more sensitive, the cell surface binding correlates better with neutralization, perhaps because it uses native S rather than stabilized S-2P. Both assays showed that all convalescent individuals had antibodies against the homologous S, as well as cross-reactive antibodies to S from other variants, although with some divergence when serum titers are low. The WA1-infected individuals showed a significant reduction in antibody titers binding to Omicron BA.1 S expressed on the cell surface (Fig. 1A) and to BA.1, BA.3 and BA.5 stabilized trimer S-2P, as well as Beta, Gamma, Omicron BA.1 and BA.2 RBD (Supplementary Fig. 1A); decreases in binding to other variants were not statistically significant. The Beta-infected individuals exhibited the highest titers against Beta S and significantly reduced titers against D614G, Delta and Omicron BA.1 S on the cell surface S-binding assay (Fig. 1A), while MSD-ECLIA data revealed reduced binding to Omicron BA.1, BA.3 and BA.5 S-2P, as well as Omicron BA.1 and BA.2 RBD (Supplementary Fig. 1A). The Gamma-infected individuals showed the least variation in antibody binding titers across the different variants (Fig. 1A), with significantly reduced titers against Omicron BA.1 and BA.5 S-2P (Supplementary Fig 1A). Consistent with previous reports[16,17], variant-infected individuals recognized WA1 RBD at similar levels as the homologous RBD (Supplementary Fig. 1A). Individuals with the highest serum binding titers (SAV1, SAV3 and A49) could cross-neutralize WA1, Beta and Gamma, and showed lower potency against Delta and Omicron BA.1 and BA.2 variants (Fig. 1B). Other individuals completely lost neutralization against Delta and Omicron variants, except for SAV11 who retained a low neutralization titer against Omicron BA.2 (Fig. 1B).

### B and T cell epitopes

We next used a surface plasmon resonance (SPR)-based competition assay[18,19] to characterize epitopes targeted by serum antibodies (Supplementary Fig. 1B). Notably, when the binding activity of each serum was characterized against the homologous S, the patterns of reactivity were comparable between individuals infected either with WA1, Beta or Gamma (Fig. 1C), revealing a conserved immunodominance hierarchy across variants, despite antigenic changes. Likewise, there were no differences in competition at each epitope when sera from Beta- or Gamma-infected individuals were mapped against WA1, Beta, Gamma or Delta S (Supplementary Fig. 1, C, D).

We evaluated the ability of T cells elicited by Beta and Gamma infections to recognize WA1 S peptides (Supplementary Dataset 1) by measuring upregulation of CD69 and CD154 on CD4 T cells, and production of IFN-γ, TNF, or IL-2 by CD8 T cells (Fig. 1D and Supplementary Fig. 2). CD4 and CD8 T cell responses to WA1 S peptides were similar in Beta- and Gamma-infected individuals compared to WA1-infected individuals (Fig. 1E). Variation was similar to that seen in IG responses (Fig. 1A), with one outlier in the CD8 responses. When stimulated with selected peptides covering only regions containing substitutions in each variant (Supplementary Dataset 2), CD4 and CD8 T cell responses were minimal, suggesting that the substituted residues are not included within immunodominant T cell epitopes (Fig. 1E).

### Isolation and characterization of cross-reactive antibodies

The three individuals in our cohort with the highest binding titers (Fig. 1A) were selected for in-depth characterization of the antibody repertoire and identification of mAb binding patterns. We developed a method for rapid assembly, transfection, and production of immunoglobulins (abbreviated to RATP-Ig) from single-sorted B cells. RATP-Ig relies on 5'-RACE and high-fidelity DNA assembly to produce recombinant heavy and light chain-expressing linear DNA cassettes, which can be directly transfected into 96-well microtiter mammalian cell cultures. Resulting culture supernatants containing the expressed mAbs can then be tested for functionality (Supplementary Fig. 3). We sorted cross-reactive WA1+Beta+ B cells (Supplementary Fig. 4, A–C) from the three selected individuals, resulting in a total of 509 single cells for analysis (Fig. 2A). We recovered paired heavy and light chain sequences from 355 (70%) cells (Fig. 2A). In parallel, we screened the RATP-Ig supernatants by ELISA for binding to S-2P, RBD, and NTD derived from each of WA1, Beta, Gamma, and Delta variants, as well as S-2P from the Omicron variant (BA.1). IgG binding at least one antigen was produced in 255 wells (50%) containing a B cell (Fig. 2A, B). All three individuals yielded high levels of cross-reactive antibodies to S, NTD, and RBD (Fig. 2B and Supplementary Dataset 3–5). Antibodies isolated from Beta-infected individuals SAV1 and SAV3 showed similar binding profiles, being dominated by cross-reactive mAbs among WA1, Beta, Gamma, and Delta variants. About half of these antibody populations comprised S-2P-only binding antibodies, with lower proportions binding NTD or RBD epitopes (Fig. 2B). From Gamma-infected individual A49, we recovered a population of mAbs that was dominated by RBD binders. While most antibodies isolated from individual A49 were also cross-reactive, we isolated a large number of mAbs whose epitope specificity we deemed indeterminate, appearing to bind both RBD and NTD (Fig. 2B and Supplementary Dataset 5), perhaps due to high-background ELISA signal. We also note that only one neutralizing antibody was recovered from A49, despite high serum binding and neutralization (Fig. 1A, B). This could perhaps be due to using two heterologous variants (WA1 and Beta) to isolate cross-reactive cells from this individual.

We next performed WA1 and Omicron pseudovirus neutralization screening for all supernatants at a 4- or 6-fold dilution. This assay identified 7, 6, and 1 antibodies neutralizing WA1 from individuals SAV1, SAV3, and A49, respectively (Fig. 2C). For most antibodies,

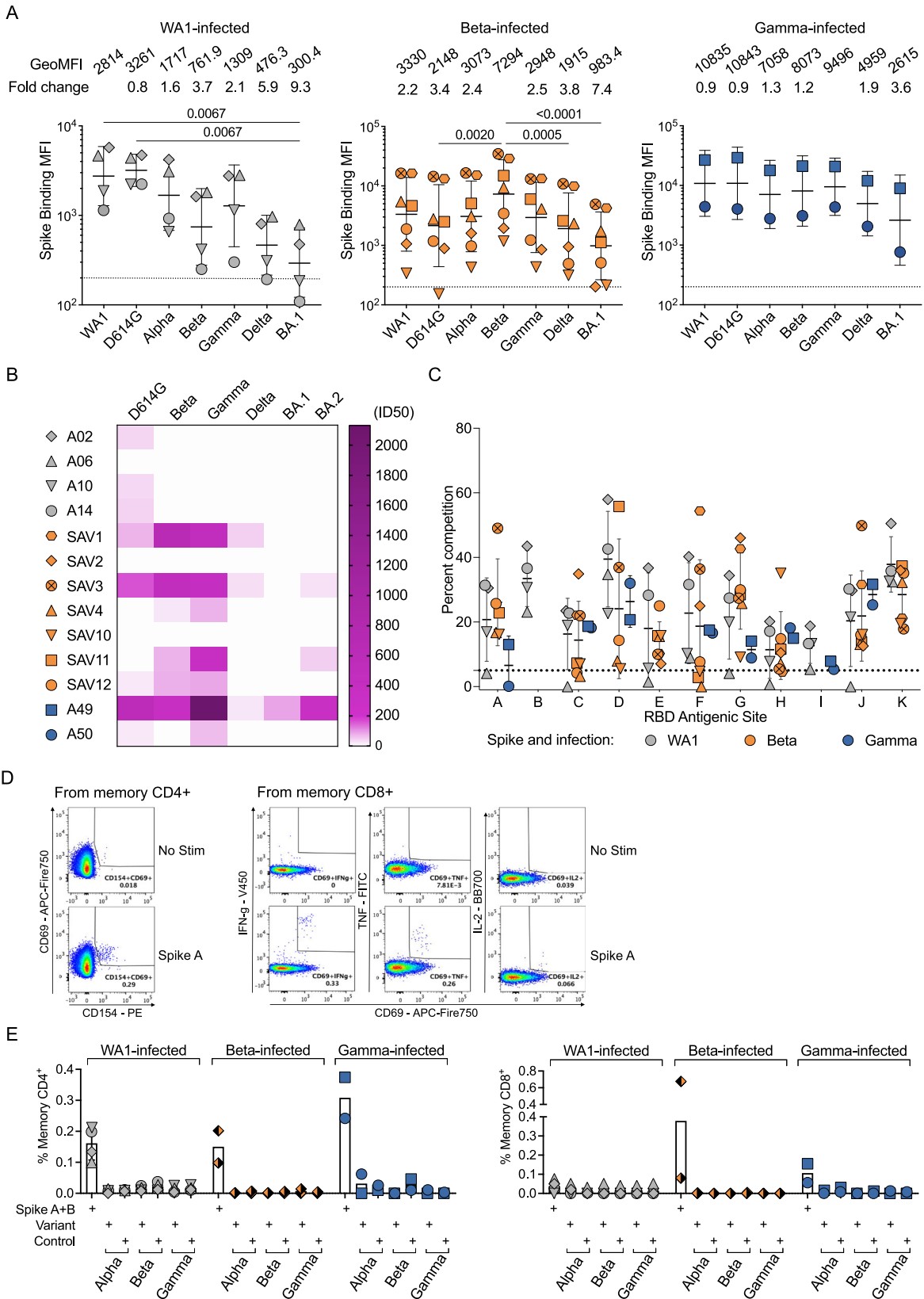

neutralization ability was diminished when tested against Omicron pseudovirus. Only three antibodies (SAV1-44.1, SAV3-4.2 and SAV3-4.3) maintained greater than 50% Omicron pseudovirus neutralization at 4- or 6-fold dilution (Fig. 2C). Neutralizing antibodies were predominately cross-reactive and RBD-specific, except for two (SAV1-159.1 and SAV3-11.1) which bound to S-2P only and a single (A49-14.1) NTD-

specific antibody (Fig. 2C). RBD-specific neutralizing antibodies were also the most potent of those isolated, with 6/12 neutralizing >90% of pseudovirus at 4-fold dilution. We validated the RATP-Ig results by selecting seven antibodies for heavy and light chain synthesis and expression and found RATP-Ig screening to be reliably predictive of mAb functionality, with 80/91 (88%) of functional interactions being

**Fig. 1 | Homologous and cross-reactive antibodies induced by WA1 and variant infections. A** Antibody binding titers against multiple variants assessed by cell surface S-binding assay. The data are shown as geometric mean with geometric standard deviation. Geometric means of median fluorescence intensity (GeoMFI) are shown on top. GeoMFI for homologous titers were used as reference to calculate fold-change in GeoMFI for the other variants. Statistical significance was determined by Friedman's test followed by Dunn's test with correction for multiple comparisons. Separate serum aliquots were used to test binding of each variant, and the results represent a single measurement per sample. **B** Heatmap showing neutralizing antibody titers (reciprocal 50% inhibitory dilution, ID50) for each individual ($N = 4$ donors in WA1-infected, 7 donors in Beta-infected, and 2 donors in Gamma-infected Supplementary Table 1) labeled on the left against each variant indicated on the top. The results represent a single measurement per sample. **C** Epitope mapping on homologous Spike by competition assay using surface plasmon resonance. Antibody CB6 (RBD-B epitope) does not bind to Beta and Gamma Spike; LY-COV555 (RBD-E epitope) does not bind to Gamma Spike; A19-30.1 (RBD-I epitope) does not bind to Beta Spike; and CR3022 (RBD-K epitope) does not bind to Gamma Spike; therefore, competition was not measured at these sites. Measurements for each individual were performed in duplicate and averaged. WA1-infected samples mapped on WA1 Spike are shown in grey, Beta-infected samples mapped on Beta Spike in orange, and Gamma-infected donors mapped on Gamma Spike in blue. Graphs show the mean for each epitope and group with standard deviation. Statistical analysis was performed using an unpaired, two-tailed *t*-test. **D** Representative FACS plots for each condition of no stim or Spike A-stimulated CD4 + (left) and CD8 + (right) memory T cells. **E** CD4 (left) and CD8 (right) T cell responses to WA1 Spike peptide pools A + B, selected pools containing altered variant peptides and control pool containing correspondent peptides for each variant pool. Results represent a single measurement per sample. Graphs show the mean for each condition with standard deviation. For **A**, **C** and **E**, symbol shapes indicate each donor, as shown in legend on panel **B**. For panel **E** only, the Beta-infected donors are from a different cohort. Source data are provided as a Source Data file.

reproduced (Supplementary Fig. 5). In summary, we found that primary infection with Beta or Gamma variants elicited similar cross-reactive B-cell responses, at single-cell resolution, targeting diverse SARS-CoV-2 epitopes.

We used SONAR[20] to identify clonally related sequences in each donor using a criteria of matching V gene and 80% CDR3 nucleotide identity in both heavy and light chain (Supplementary Dataset 6). All three individuals had polyclonal antigen-specific repertoires (Fig. 2D). Interestingly, clones observed in more than one cell were more frequently derived from *IGHV1-69* (9/33 expanded clones) and *IGHV3-30*-subfamily genes (15/33 expanded clones) than were singleton clones (30/388 and 70/388 clones, respectively). SHM levels were the same between expanded and unexpanded clones, while 12/14 neutralizers were singletons in our data. However, the limited sampling here would obscure modest differences in levels of expansion between clones.

SAV3 and A49 had highly expanded clones matching a widely reported public clone using *IGHV1-69* and *IGKV3-11*[9,21–24]. Members of this public clone were also recovered from SAV1, although they were not greatly expanded. RATP-Ig ELISA data indicated that these antibodies bound a non-RBD, non-NTD epitope on S-2P, consistent with available data for previously described members of this public clone. Notably, all but one of the antibodies we recovered from this public clone bound to Delta S-2P, and 11/17 also bound to Omicron S-2P. In addition, most antibodies from this public clone have been reported to bind SARS-CoV-1[9,21–24], and one, mAb-123[22], weakly binds endemic human coronaviruses HKU1 and 229E. We also found two antibodies, SAV1-109.1 and SAV1-168.1, with a YYDRxG motif in CDR H3 that can target the epitope of mAb CR3022 on RBD and produce broad and potent neutralization of a variety of sarbecoviruses[25]. While SAV1-168.1 was cross-reactive but non-neutralizing (Supplementary Dataset 3), SAV1-109.1 showed good neutralization potency and bound to WA1, Beta, Gamma and Delta, but not Omicron (Fig. 2C). Overall, 185 (90%) of the 206 WA1/Beta cross-binding mAbs also bound Delta, while only 109 (53%) of those mAbs also bound Omicron (Supplementary Dataset 3–5).

### In-depth characterization of antigen-specific B cells
To investigate possible differences in targeting of domains outside of RBD, we further examined epitope specificities by flow cytometry (Supplementary Fig. 4B, D). As expected, the frequency of antigen-specific cells generally correlated with serum binding titers, and cells capable of binding to heterologous variants were typically less frequent than those binding the infecting variant (Fig. 3A). In addition, both Beta- and Gamma-infected individuals showed higher frequencies of NTD-binding B cells against the homologous virus when compared to WA1-infected individuals (Fig. 3B).

We sorted 847, 5806 and 494 antigen-specific single cells from the WA1-, Beta- and Gamma-infected groups, respectively (Supplementary

Figs. 4E and Supplementary Table 2). We then generated libraries using the 10x Genomics Chromium platform and recovered a total of 162, 319, and 107 paired heavy and light chain sequences from the WA1-, Beta-, and Gamma-infected groups, respectively (Supplementary Table 2). As observed in the sequences identified via RATP-Ig, all three SARS-CoV-2-specific IG repertoires showed little clonal expansion (Supplementary Fig. 6). We then combined these data with the sequences generated by RATP-Ig for downstream analysis. Antigen-specific V gene usage was highly similar across all three infection types (Fig. 3C and Supplementary Table 3), with differences noted for *IGHV1-46*, *IGLV1-44*, and *IGLV1-47* (Supplementary Fig. 7A, Supplementary Table 3). However, when we compared these antigen-specific repertoires to the total memory B cell repertoire in pre-pandemic controls[26], we observed significant enrichment for *IGHV3-30* and depletion of *IGHV1-18* and *IGKV1-27* in the WA1- and Gamma-infected groups and *IGHV4-30-2* in the WA1- and Beta-infected groups (Fig. 3C, Supplementary Fig. 7B, and Supplementary Table 3). Although these changes from the control repertoires were not observed in the Beta-infected cohort, this is likely due to low sampling of SAV2 and SAV10 (Supplementary Table 2). Overall, IG V gene usage is markedly consistent across the responses to all SARS-CoV-2 variants we investigated.

Recent studies have shown that Y501-dependent mAbs derived from *IGHV4-39* and related genes are overrepresented among neutralizing antibodies isolated from Beta-infected individuals[27,28]. We, therefore, analyzed the observed frequency of *IGHV4-30*, *IGHV4-31*, *IGHV4-39*, and *IGHV4-61* among Beta- and Gamma-binding B cells sequenced from 10x Genomics libraries but found no significant differences based on infecting variant (Fig. 3D). RATP-Ig-derived sequences, including those we identified as neutralizing, are not expected to be enriched for this class of antibodies, as they were specifically sorted for cross-binding to WA1, which does not contain residue Y501. Furthermore, we compared the frequency of sequences using these germline genes for WA1- versus Beta-binding B cells among Beta-infected individuals (again excluding cross-reactive B cells isolated by RATP-Ig), and again found no difference in usage (Fig. 3E). The lack of observed enrichment for these genes is likely due to the fact that neutralizing antibodies comprise only a small fraction of the antigen-specific binding repertoire[9,29], with the latter remaining highly conserved across individuals infected with different variants.

We next investigated SHM levels in these repertoires. The median $V_H$ SHM levels among individuals was 0.3–6.6% in $V_H$ and 0.0–3.0% in $V_L$, compared to 6.7% and 2.4%, respectively, in the control repertoires. We then further examined SHM by both infecting variant and the probes used to isolate each cell. We found no differences in SHM in single probe-binding repertoires for either WA1- or Gamma-infected individuals (Fig. 4). Surprisingly, cross-reactive (WA1 and Beta) cells sorted for RATP-Ig had lower SHM

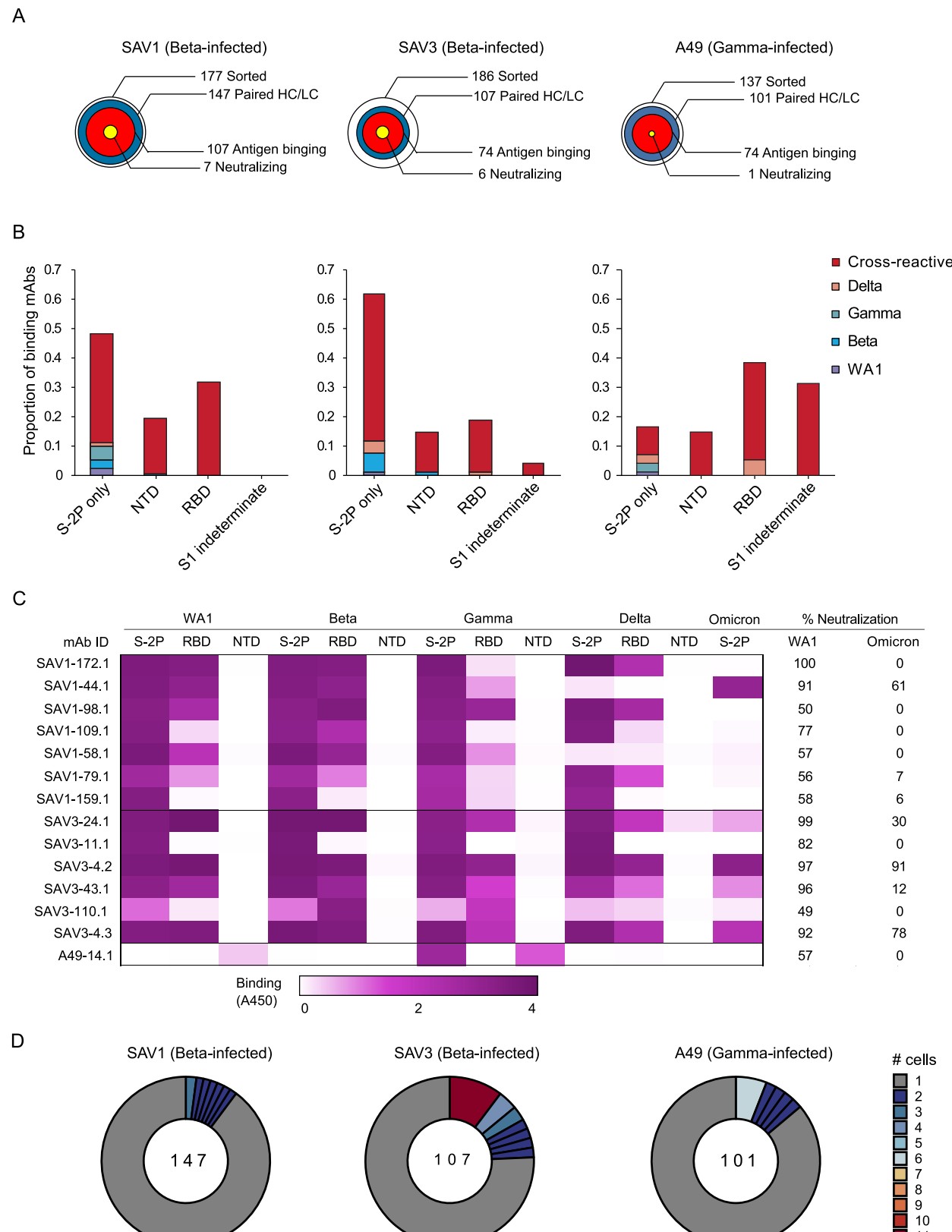

Binding (A450)

than the single probe-binding repertoires sorted for 10x Genomics and sequencing. This may suggest that Beta S-2P is a better immunogen, capable of stimulating naïve B cells that require less SHM to gain cross-reactivity. Moreover, single probe-binding Beta-specific B cells from Beta-infected individuals had significantly higher SHM (median of 4.9% in $V_H$ and 2.7% in $V_L$) compared to single probe WA1-

binding cells from the same individuals (2.1% and 0.8%, respectively) (Fig. 4). Other studies have also suggested the possibility that the immune response to Beta may be somewhat distinct from that against other SARS-CoV-2 variants, with neutralization appearing to wane more slowly and rising to higher levels after additional vaccine doses[18,19]. Overall, the low levels of SHM across all the SARS-CoV-2-

**Fig. 2 | Functional Characterization of RATP-Ig-Isolated mAbs. A** RATP-Ig screening overviews for three individuals, represented as bullseyes. The area of each circle is proportional to the number of antibodies indicated on the figure. **B, C** Supernatants were screened for antigen-specific binding by single-point ELISA for WA1, Beta, Gamma, and Delta S-2P, RBD, and NTD, as well as Omicron S-2P. Results were obtained by a single measurement per mAb. Source data are provided as a Source Data file. **B** Each panel represents the exact proportion of mAbs isolated from a single individual indicated in **A**, where cross-reactive (red bars) represent any combination of variants. **C** ELISA heatmap reported as absorbance at 450 nm (not quantitative) and neutralization screening (right columns) of isolated mAbs at 4- or 6-fold supernatant dilutions using a pseudovirus luciferase reporter assay for WA1 and Omicron, reported as % virus neutralized derived from reduction in luminescence. Measurements for each mAb were performed in triplicate and averaged. **D** Clonal expansion in each individual. Expanded clones are colored by the number of cells in each clone as shown on the right; singleton clones are shown in gray. The numbers of total clones sequenced are shown in the middle of each circle.

specific B cells that we isolated is consistent with prior reports[13,23,29] and further demonstrates that the human immune system can readily generate antibodies capable of cross-binding multiple variants, regardless of infecting variant.

## Public B cell clones

We next identified public clones in the SARS-CoV-2-specific repertoires elicited after infection with different variants. Relative to true biological clones, we defined public clones differently in two ways. CDR3 identity was calculated in terms of amino acids, instead of nucleotides, and only the heavy chain was considered. Both changes reflect an emphasis on features likely to indicate similar functionality, which can be broader than those reflecting a shared V(D)J rearrangement event.

In total, 16 public clones were identified from 11 of the 13 infected individuals distributed across infection with all three variants (Fig. 5A). Notably, public clones for which data are available bound to Delta S-2P, and a subset of antibodies from the two most abundant public clones also bound to Omicron S-2P. One public clone, found in 5 individuals, uses *IGHV4-59* with a short, strongly conserved, 6 amino acid CDR H3 and *IGKV3-20* (Fig. 5A, B). Antibodies matching the signature of public clone 1 have been previously described to bind the S2 domain of S and are generally cross-reactive with SARS-CoV-1[9,21,29]. Indeed, one member of this public clone was isolated from an individual infected with SARS-CoV-1[21]. This suggests that the convergent immune responses we observe may not be elicited only by variants of a single virus such as SARS-CoV-2 but can even extend to a broader range of related viruses.

Public clones 2 and 3 both use the same heavy and light chain germline genes with the same CDR H3 and L3 lengths, though they fall outside of the 80% amino acid identity threshold. Combining sequences from both public clones revealed a strongly conserved IGHD3-22-encoded YDSSGY motif at positions 6-11 of CDR H3 (Fig. 5C). Strikingly, this is the same D gene implicated in targeting a Class IV RBD epitope[25] although public clones 2 and 3 instead target an epitope in S2 and appear to be restricted to *IGHV1-69* and *IGKV3-11* V genes. We also observed the repeated use of *IGHV3-30* with a 14 amino acid CDR H3 in six public clones which together comprise 35 cells from 8 different individuals. When we combined CDR H3 sequences from all 6 public clones in this group, we found an IGHD1-26-derived small-G-polar-Y-aromatic motif spanning positions 5–9 of CDR H3 (Fig. 5D). A large number of antibodies matching this signature have been previously described[7,9,21–24,29,30]. The repeated observation of these closely related public clones in multiple individuals across all studied infection types further demonstrates the extraordinary convergence of the immune response to diverse variants.

We identified only one public clone, 12, that we were able to verify bound to either RBD or NTD, although public clones 13 and 14 also have highly similar V genes and CDR lengths (Fig. 5A, E). Two previously reported antibodies, WRAIR-2038[31] and COV-2307[23], match the signature of these public clones and are also confirmed to bind NTD. The identification of a cross-reactive public clone is remarkable given deletions in Beta that disrupt the main NTD supersite for neutralizing antibodies[32]. This again highlights the capacity of the adaptive immune response to find consistent ways to target the SARS-CoV-2 virus, despite substitutions selected for their ability to disguise the targets.

## Discussion

A deep understanding of the IG repertoires that mediate cross-protective responses to SARS-CoV-2 after infection or vaccination will be critical for guiding therapeutic approaches to future variants as the virus continues to evolve. In this study, we used rapid mAb production and functional analysis, and single-cell IG sequencing to conduct an in-depth, unbiased characterization of total antigen-specific B cell responses against multiple SARS-CoV-2 variants, including Delta and Omicron, in people infected with the ancestral WA1, Beta, or Gamma. Our principal findings were: (1) infection with any of the "older" variants consistently elicited substantial numbers of antibodies capable of cross-binding even to the more recent antigenically divergent variants Delta and Omicron; (2) infection with any of these variants elicited antibodies targeting the same immunodominant epitopes in RBD; (3) antigen-specific memory B cells elicited by SARS-CoV-2 are polyclonal and use similar patterns of heavy and light chain V genes, irrespective of the infecting variant; and 4) public clones and other cross-reactive antibodies are common among responses to all infecting variants. Our results demonstrate a fundamentally convergent humoral immune response across different SARS-CoV-2 variants that cross-bind even to antigenically distant ones such as Delta and Omicron.

To date, most analyses of SARS-CoV-2-specific B cells have focused on neutralizing antibodies with potential therapeutic applications. Those which have investigated the total binding repertoire have used samples from people infected with the ancestral WA1 variant[7,10]; here, we extend such analysis to individuals infected with the antigenically distinct Beta and Gamma variants and show that antibodies capable of binding to multiple variants are common. Indeed, while the strength of cross-neutralization depends on the antigenic distance from the infecting variant[33], we found that most WA1-Beta cross-binding antibodies can also bind to a later, more divergent, variants such as Delta, and approximately half can additionally bind Omicron.

Furthermore, we observed that the hierarchy of immunodominant epitopes targeted on these variants remains unchanged. While a recent report found that Beta-infection was less likely to elicit antibodies contacting S residue F456 than WA1-infection[34], we found no changes in targeting of the RBD-A epitope, which includes this residue. Interestingly, even though the immunodominance of binding epitopes is known to be consistent in response to WA1, Beta, or Omicron mRNA immunization[18,19], recent reports have found that infection with an Omicron subvariant after vaccination can shift the epitope landscape compared to vaccination alone[35,36]. This likely reflects the effect of imprinting by consecutive exposures to closely related antigens[37], although differences in the primary response to Omicron variants cannot yet be ruled out. For earlier variants, at least, we demonstrate here similar patterns of immunodominance after variant infection, a phenomenon that may help explain the continued efficacy of vaccines based on ancestral variants.

In addition to concordant epitope targeting, we also found consistent IG V gene usage in the antibody response to all three variants we investigated. Our findings highlight the difference between the neutralizing antibody repertoires investigated previously compared to the total binding repertoires examined here, emphasizing the insights to be gleaned by taking a broader perspective. Thus, while many of the

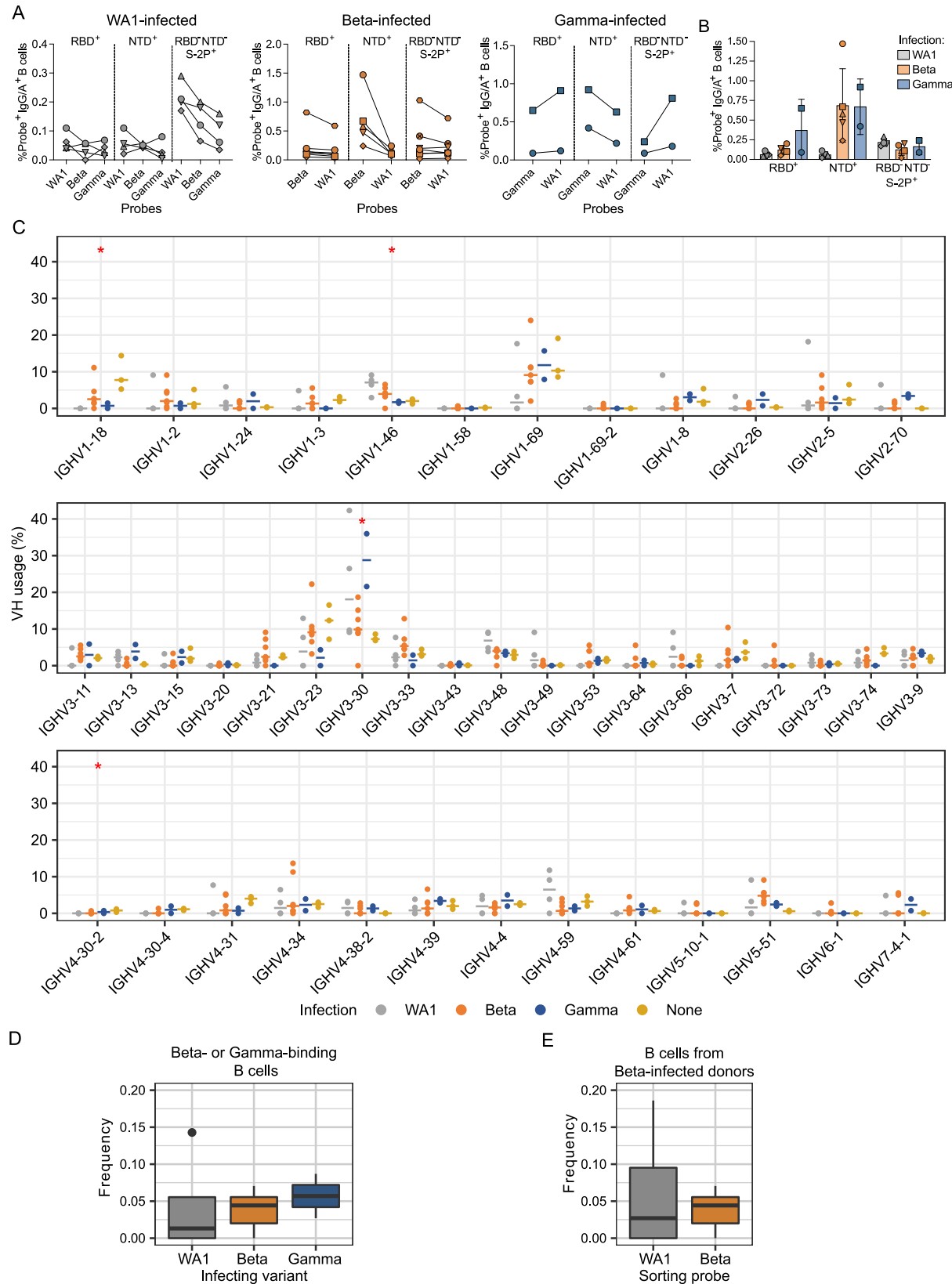

variant-induced public clones that were cross-reactive with all three variants, as well as Delta and sometimes Omicron, appear to be non-neutralizing and S2 domain-binding, the breadth and ready elicitation may be important for Fc-dependent functions[31,38]. Therefore, public clones stimulated by one variant could play a protective role against later variants, even when neutralizing antibodies are less effective.

Overall, more than 8% of the cells that we sequenced belong to a public clone, highlighting again the extraordinary convergence of the anti-body response across antigenically distinct variants of SARS-CoV-2. Importantly, even when sequence homology fell below the threshold to define clones as public, we found conserved motifs which are likely to drive functional convergence consistent with recent evidence that

**Fig. 3 | Anti-SARS-CoV-2 IG repertoires. A** Frequencies of probe+ IgG or IgA B cells sorted for IG repertoire analysis. Symbol shapes indicate each donor as on legend for panel 1B. Results were obtained by a single measurement per sample. **B** Proportion of probe+ B cells binding to each domain. Bars represent means with standard deviations. Source data for **A** and **B** are provided as a Source Data file. **C** SARS-CoV-2-specific VH repertoire analysis by infecting variant WA1, Beta and Gamma shown in grey, orange and blue, respectively, with data from pre-pandemic controls in yellow. The x-axis shows all germline genes used; y-axis represents percent of individual gene usage. Horizontal lines show the median of each group for each gene. Red stars indicate genes with at least one significant difference between groups based on a Kruskal–Wallis test; pairwise comparisons using the Dunn test with correction for multiple testing are in Supplementary Table 3. *n* = 133,

737, 190, and ~7 × 10⁸ heavy chains from 4, 7, 2, and 3 individuals, for WA1-infected, Beta-infected, Gamma-infected, and historical controls, respectively. Combined frequency of VH genes capable of giving rise to stereotypical Y501-dependent antibodies (*IGHV4-30*, *IGHV4-31*, *IGHV4-39*, and *IGHV4-61*) in **D** Beta- or Gamma-binding B cells from individuals infected with each variant or **E** B cells from Beta-infected individuals sorted with either WA1- or Beta-derived probes. For both **D** and **E**, the boxes show the interquartile range, with the median marked as heavy horizontal band. Whiskers represent the highest (lowest) datapoint within 1.5 times the interquartile range of the 75th (25th) percentile. In panel **D**, *n* = 83, 349, and 60 cells from 4, 5, and 2 individuals for WA1-, Beta-, and Gamma-infection, respectively. In panel **E**, *n* = 111 and 349 cells from 5 individuals which bound to WA1 and Beta probes, respectively.

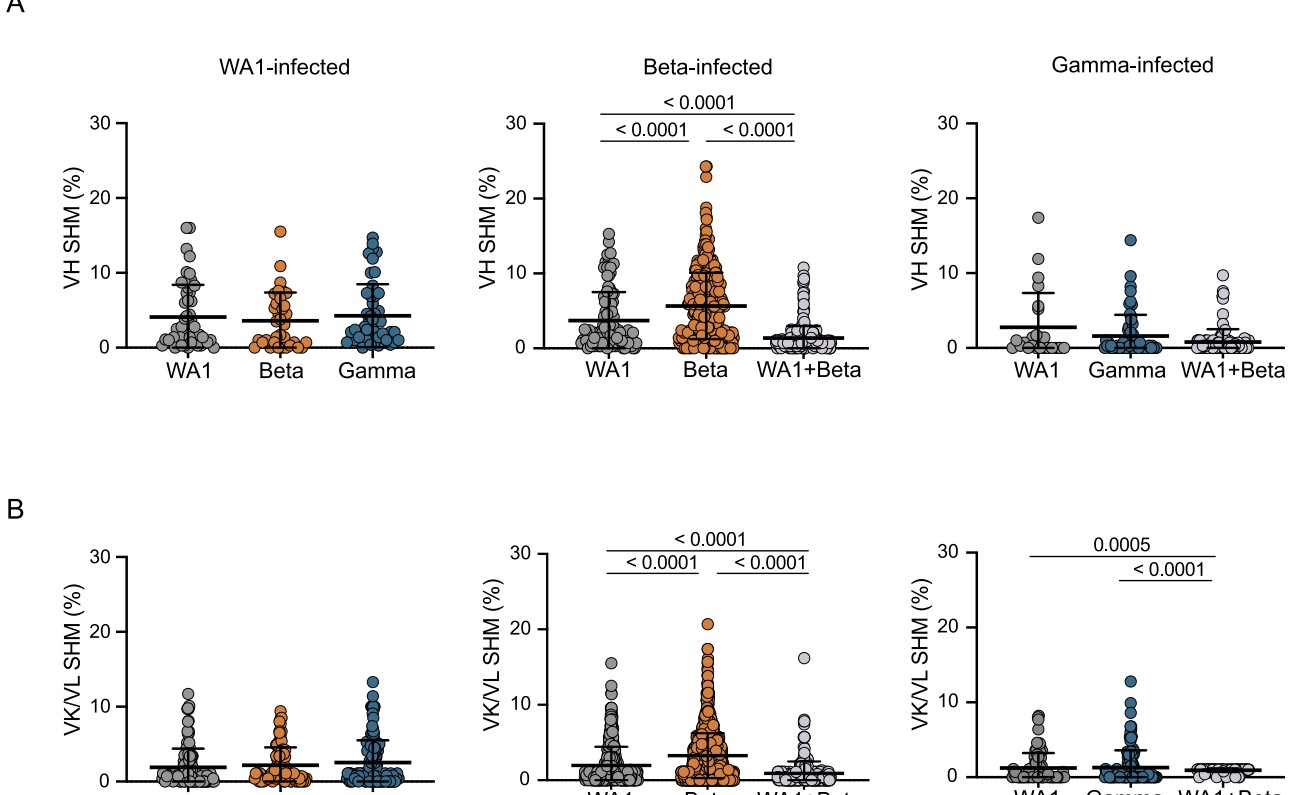

**Fig. 4 | Somatic hypermutation (SHM) levels of SARS-CoV-2 specific B cells (unpaired sequences). SHM percent in variable heavy (VH). A** or variable kappa/lambda ($V_K/V_L$) **B** regions. Error bars indicate the average percentage of nucleotide substitutions +/- standard deviation. Statistical significance was determined using a two-tailed Mann-Whitney U test. Source data are provided as a Source Data file. In panel **A** *n* = 50, 34, and 49 heavy chains, respectively, for WA1-infected individuals;

111, 349, and 277 heavy chains for Beta-infected; and 24, 60, and 106 heavy chains for Gamma-infected. In panel **B** *n* = 125, 88, and 136 light chains, respectively, for WA1-infected; 380, 1157, and 289 light chains for Beta-infected; and 81, 112, and 109 light chains for Gamma-infected. These numbers included heavy and light chains from cells for which the cognate light or heavy chain was not recovered.

antibodies may target overlapping epitopes using comparable binding conformations in the absence of convergent V genes[39]. Together, these findings further highlight the capability of the human immune system to respond to SARS-CoV-2 in a manner that is largely conserved yet at the same time tolerant of differences between variants.

In summary, our data reveal marked convergence that defines multiple aspects of the humoral immune response to different SARS-CoV-2 variants. This phenomenon comprises convergent V-gene usage and epitope specificities elicited by primary exposure to SARS-CoV-2 variants, including a substantial proportion of public clones and cross-binding B cells. This suggests the existence of immunological constraints guiding the response to related viruses, even in the face of substantial antigenic divergence, and may explain how first-generation

vaccine designs using the ancestral S protein sequence have generally proven equally as protective against severe disease compared to updated vaccines matched to recent variants[40,41].

Our study is limited by sampling of paired heavy and light chain sequences from fewer than 1,000 SARS-CoV-2-specific B cells across 13 individuals. This scale is small in comparison to bulk IG sequencing studies and even a few single-cell studies. In addition, we do not know the genetic background of the individuals in this study, and so cannot address whether specific allelic variants of highly polymorphic $V_H$ genes might be important for the public clones we observe. We are also limited in our ability to make functional repertoire comparisons due to varied sorting strategies and differences in functional assays used to assess isolated mAbs. Moreover, our cohort was sampled only

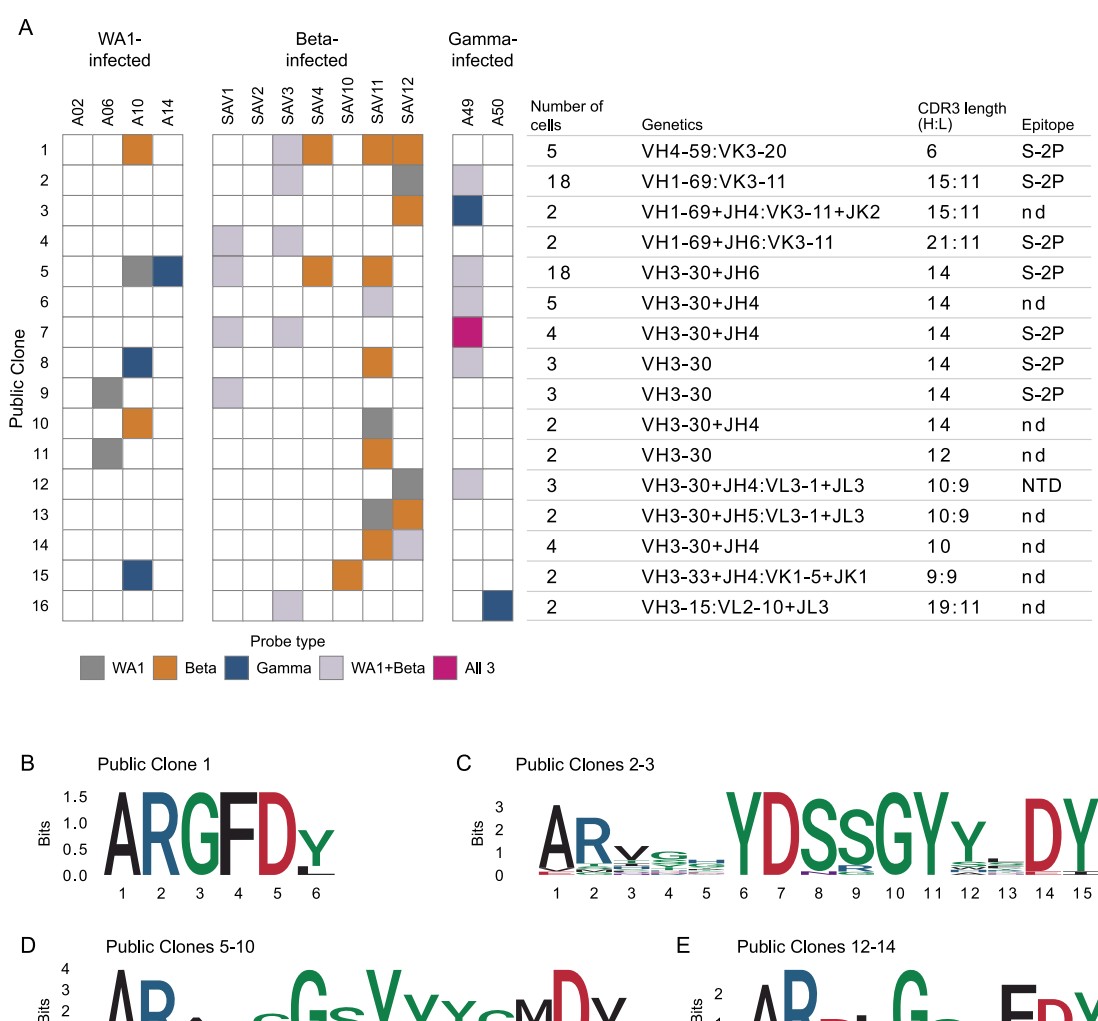

**Fig. 5 | Public and cross-reactive clones. A** Sixteen public clones were identified. Public clones are numbered 1-16 by row, as shown on the far left. Each column of boxes in the middle panel represents a single individual, as labeled at top, and is colored by probe(s) used, as shown at bottom. Right panel shows additional information about each public clone. Light chain information is provided after a colon if a consistent signature was found. Epitopes are inferred from ELISA of RATP-Ig supernatants of at least 1 public clone member; nd, not determined. **B** CDR H3 logogram for the top public clone, found in 5 of 13 individuals. **C–E** Combined CDR H3 logograms for 2 public clones using *IGHV1-69* and *IGKV3-11* with a 15 amino acid CDR H3 length. **D** 6 public clones using *IGHV3-30* with a 14 amino acid CDR H3 length. **E** 3 public clones using *IGHV3-30* with a 10 amino acid CDR H3 length.

at a single time point early in convalescence and included only one individual with high serum neutralization titers. It will be important to verify that our findings extend to later time points when the antibody repertoire has matured. In addition, while Beta and Gamma are antigenically distinct from WA1, they only represent a small portion of the SARS-CoV-2 antigenic map[42]. Further studies are needed to examine the response elicited by more antigenically divergent SARS-CoV-2 variants such as Delta and Omicron.

## Methods
### Study design
We selected 13 convalescent individuals that had experienced symptomatic Covid-19 infection with either WA1 virus or the Beta or Gamma variants. Serum, plasma and PBMC were isolated at each respective clinical center. The selection of individuals was based on the availability of samples collected at similar time-points (between 17 and 38 days after symptoms onset), rather than the severity of disease or neutralizing antibody titers (Supplementary Table 1).

Seven individuals were infected with the Beta variant and recruited at the Sheba Medical Center, Tel HaShomer, Israel. Because of limited sample availability, two additional Beta-infected individuals were recruited at the Vaccine Research Center (VRC) and used for T cell analyses. Two individuals were infected with the Gamma variant and recruited at the University of Minnesota Hospital, USA. Infections with Beta and Gamma variants were confirmed by sequencing. The samples from four WA1-infected individuals, collected early in the pandemic prior to the emergence of variants, as well as the two additional beta-infected individuals used for T cell analysis were collected under the VRC, National Institute of Allergy and Infectious Diseases (NIAID), National Institutes of Health's protocol VRC 200 (NCT00067054) in compliance with the NIH Institutional Review Board (IRB) approved protocol and procedures. All subjects met protocol eligibility criteria and agreed to participate in the study by signing the NIH IRB approved informed consent. Research studies with these samples were conducted by protecting the rights and privacy of the study participants. All participants provided

informed consent in accordance with protocols approved by the respective IRB and the Helsinki Declaration.

## Serology

Antibody binding was measured by 10-plex Meso Scale Discovery Electrochemiluminescence immunoassay (MSD-ECLIA) as previously described[4]. Cell-surface S binding was assessed as previously described[4]. Serum neutralization titers for either WA1-D614G, Beta, Gamma or Delta pseudotyped virus particles were obtained as previously described[4].

## Antigen-specific ELISA

Reacti-Bind 96-well polystyrene plates (Pierce) were coated with 100 μl of affinity purified goat anti-human IgG Fc (Rockland) at 1:20,000 in PBS, or 2 μg/ml SARS-CoV-2 recombinant protein in PBS overnight at 4 °C. Plates were washed in PBS-T (500 ml 10XPBS + 0.05% Tween-20 + 4.5 L H2O) and blocked for 1 h at 37 °C with 200 μl/well of B3T buffer: 8.8 g/liter NaCl, 7.87 g/liter Tris-HCl, 334.7 mg/liter EDTA, 20 g BSA Fraction V, 33.3 ml/liter fetal calf serum, 666 ml/liter Tween-20, and 0.02% Thimerosal, pH 7.4). Diluted antibody samples were applied and incubated 1 h at 37 °C followed by 6 washes with PBS-T; plates were then incubated with HRP-conjugated anti-human IgG (Jackson ImmunoResearch) diluted 1:10,000 in B3T buffer for 1 h at 37 °C. After 6 washes with PBS-T, SureBlue TMB Substrate (KPL) was added, incubated for 10 min, and the reaction was stopped with 1 N H2SO4 before measuring optical densities at 450 nm (Molecular Devices, SpectraMax using SoftMax Pro 5 software). For single-point assays, supernatants from transfected cells were diluted 1:10 in B3T and added to the blocked plates. ELISA signals were considered positive if they were greater than or equal to 2X the average of the blank wells of the plate.

## Pseudovirus neutralization assay

SARS-CoV-2 spike pseudotyped lentiviruses were produced by co-transfection of 293 T cells with plasmids encoding the lentiviral packaging and luciferase reporter, a human transmembrane protease serine 2 (TMPRSS2), and SARS-CoV-2 S genes using Lipofectamine 3000 transfection reagent (ThermoFisher, CA)[15,43]. Forty-eight hours after transfection, supernatants containing pseudoviral particles were harvested, filtered and frozen. For neutralization assay two dilutions of the transfection supernatants (2- or 3-fold) were mixed with equal volume of titrated pseudovirus (final dilution 4x or 6x), incubated for 45 min at 37 °C and added to pre-seeded 293 flpin-TMPRSS2-ACE2 cells (made by Adrian Creanga, VRC, NIH) in triplicate in 96-well white/black Isoplates (Perkin Elmer). Following 2 hours of incubation, wells were replenished with 150 μL of fresh medium. Cells were lysed 72 h later and luciferase activity (relative light unit, RLU) was measured. Percent neutralization was calculated relative to pseudovirus-only wells. Measurements for each mAb were performed in triplicate and averaged.

## Intracellular cytokine staining

The T cell staining panel used in this study was modified from a panel developed by the laboratory of Dr. Steven De Rosa (Fred Hutchinson Cancer Research Center). Directly conjugated antibodies purchased from BD Biosciences include CD19 PE-Cy5 (Clone HIB19; cat. 302210), CD14 BB660 (Clone M0P9; cat. 624925), CD3 BUV395 (Clone UCHT1; cat. 563546), CD4 BV480 (Clone SK3; cat. 566104), CD8a BUV805 (Clone SK1; cat. 612889), CD45RA BUV496 (Clone H100; cat. 750258), CD154 PE (Clone TRAP1; cat. 555700), IFNγ V450 (Clone B27; cat. 560371 and IL-2 BB700 (Clone MQ1-17H12; cat. 566404). Antibodies from Biolegend include CD16 BV570 (Clone 3G8; cat. 302036), CD56 BV750 (Clone 5.1H11; cat. 362556), CCR7 BV605 (Clone G043H7; cat. 353244) and CD69 APC-Fire750 (Clone FN50; cat. 310946). TNF FITC (Clone Mab11; cat. 11-7349-82) and the LIVE/DEAD Fixable Blue Dead Cell Stain (cat. L34962) were purchased from Invitrogen. For dilutions of antibodies see Supplementary Table 4.

Cryopreserved PBMC were thawed into pre-warmed R10 media (RPMI 1640, 10% FBS, 2 mM L-glutamine, 100 U/ml penicillin, and 100 μg/ml streptomycin) containing DNase and rested for 1 h at 37 °C/5% CO2. For stimulation, 1–1.5 million cells were plated into 96-well V-bottom plates in 200 mL R10 and stimulated with SARS-CoV-2 peptide pools (2ug/mL for each peptide) in the presence of Brefeldin A (Sigma-Aldrich) and monensin (GolgiStop; BD Biosciences) for 6 hours at 37 °C/5%CO2. A DMSO-only condition was used to determine background responses. Following stimulation samples were stained with LIVE/DEAD Fixable Blue Dead Cell Stain for 10 min at room temperature and surface stained with titrated amounts of anti-CD19, anti-CD14, anti-CD16, anti-CD56, anti-CD4, anti-CD8, anti-CCR7 and anti-CD45RA for 20 min at room temperature. Cells were washed in FACS Buffer (PBS + 2% FBS), and fixed and permeabilized (Cytofix/Cytoperm, BD Biosciences) for 20 min at room temperature. Following fixation, cells were washed with Perm/Wash buffer (BD Biosciences) and stained intracellularly with anti-CD3, anti-CD154, anti-CD69, anti-IFNγ, anti-IL-2 and anti-TNF for 20 min at room temperature. Cells were subsequently washed with Perm/Wash buffer and fixed with 1% paraformaldehyde. Data were acquired on a modified BD FACSymphony and analyzed using FlowJo software (version 10.7.1). Cytokine frequencies were background subtracted and negative values were set to zero.

Synthetic peptides (>75% purity by HPLC; 15 amino acids in length overlapping by 11 amino acids) were synthesized by GenScript. To measure T cell responses to the full-length WA1 S glycoprotein (YP_009724390.1), two peptide pools were utilized, S pool A (peptides 1-160; residues 1-651) and S pool B (peptides 161-316; residues 641-1273) (Supplementary Dataset 1). Peptides were 15 amino acids in length and overlapped by 11 amino acids. S pool A contained peptides for both D614 and the G614 mutation. Responses to full-length S were calculated by summing the responses to both pools after background subtraction. Select peptide pools were used to measure T cell responses to mutated regions of the S glycoproteins of the Alpha, Beta and Gamma SARS-CoV-2 variants along with control pools corresponding to the same regions within the WA1 S glycoprotein (Supplementary Dataset 2).

## Epitope mapping by Surface Plasmon Resonance (SPR)

Serum epitope mapping competition assays were performed, as previously described[18,19], using the Biacore 8 K + surface plasmon resonance system (Cytiva). Anti-histidine antibody was immobilized on Series S Sensor Chip CM5 (Cytiva) through primary amine coupling using a His capture kit (Cytiva). Following this, his-tagged SARS-CoV-2 S protein containing 2 proline stabilization mutations (S-2P) was captured on the active sensor surface.

Human IgG monoclonal antibodies (mAb) used for these analyses include: B1-182, CB6, A20-29.1, A19-46.1, LY-COV555, A19-61.1, S309, A23-97.1, A19-30.1, A23-80.1, and CR3022. Either competitor or negative control mAb was injected over both active and reference surfaces. Human sera were then flowed over both active and reference sensor surfaces, at a dilution of 1:50. Following the association phase, active and reference sensor surfaces were regenerated between each analysis cycle.

Prior to analysis, sensorgrams were aligned to Y (Response Units) = 0, using Biacore 8 K Insights Evaluation Software (Cytiva), at the beginning of the serum association phase. Relative "analyte binding late" report points (RU) were collected and used to calculate percent competition (% C) using the following formula: % C = [1 − (100 * ((RU in presence of competitor mAb) / (RU in presence of negative control mAb))]. Results are reported as percent competition and statistical analysis was performed using unpaired, two-tailed t-test (Graphpad Prism v.8.3.1). All assays were performed in duplicate and averaged.

Only one of the WA1-infected individuals (A14) produced sufficiently high binding titers against Beta and Delta S to enable epitope

mapping by competition. In addition, Beta-infected donors SAV2 and SAV10 were below the lower limit of quantification for WA1 and Delta S.

## Production of antigen-specific probes

Biotinylated probes for S-2P, NTD and RBD were produced as described previously[44,45]. Briefly, single-chain Fc and AVI-tagged proteins were expressed transiently for 6 days. After harvest, the soluble proteins were purified and biotinylated in a single protein A column followed by final purification on a Superdex 200 16/600 gel filtration column. Biotinylated proteins were then conjugated to fluorescent streptavidin.

## Antigen-specific B cell sorting

PBMC vials containing approximately $10^7$ cells were thawed and stained with Live/Dead Fixable Blue Dead Cell Stain Kit (Invitrogen, cat# L23105) for 10 min at room temperature, followed by incubation for 20 min with the staining cocktail consisting of antibodies and probes. The antibodies used in the staining cocktail were: CD8-BV510 (Biolegend, clone RPA-T8, cat# 301048), CD56-BV510 (Biolegend, clone HCD56, cat# 318340), CD14-BV510 (Biolegend, clone M5E2, cat# 301842), CD16-BUV496 (BD Biosciences, clone 3G8, cat# 612944), CD3-APC-Cy7 (BD Biosciences, clone SP34-2, cat# 557757), CD19-PECy7 (Beckmann Coulter, clone J3-119, cat# IM36284), CD20 (BD Biosciences, clone 2H7, cat# 564917), IgG-FITC (BD Biosciences, clone G18-145, cat# 555786), IgA-FITC (Miltenyi Biotech, clone IS11-8E10, cat# 130-114-001) and IgM-PECF594 (BD Biosciences, clone G20-127, cat# 562539). For each variant, a set of two S probes S-2P-APC and S-2P-BUV737, in addition to RBD-BV421 and NTD-BV711 were included in the staining cocktail for flow cytometry sorting. For dilutions of antibodies see Supplementary Table 4.

For RATP-Ig, single-cells were sorted in 96-well plates containing 5 µL of TCL buffer (Qiagen) with 1% β-mercaptoethanol according to the gating strategy shown in Supplementary Fig. 3B, C. Samples sorted for 10x Genomics single-cell RNAseq were individually labelled with an oligonucleotide-linked hashing antibody (Totalseq-C, Biolegend) in addition to the staining cocktail and sorted into a single tube according to the gating strategy shown in Supplementary Fig. 3B, E. All cell sorts were performed using a BD FACSAria II instrument (BD Biosciences) with BD FACSDiva Software version 9.5.1 (BD Biosciences). Frequency of antigen-specific B cells were analyzed using FlowJo 10.8.1 (BD Biosciences).

## Monoclonal antibody isolation and characterization by RATP-Ig

**Overview.** Rapid Assembly, Transfection and Production of Immunoglobulins is a novel method we developed to rapidly isolate monoclonal antibodies from single sorted B cells without plasmid cloning. The method can be broken down into 4 steps: (1) single-cell cDNA synthesis; (2) immunoglobulin enrichment and sequencing; (3) cassette fragment synthesis; (4) and cassette assembly. Together, these steps produce a linear, double-stranded gene cassette capable of expressing full-length human, non-human primate, or mouse (using different primers) antibodies. Our current version can recover and express IgA, IgG, IgK, IgL combinations of antibodies. All antibodies, regardless of original heavy/light chain pair, are then expressed on an IgG/IgK constant region. All reactions are plate-based, and can be performed in under a week, making RATP-Ig a high-speed, high-throughput, and high-fidelity method for pairing antibody sequence to function.

**Single-cell cDNA synthesis.** Variable heavy and light chains were synthesized using a modified SMARTSeq-V4 protocol by 5' RACE. Single-cell RNA was first purified with RNAclean beads (Beckman Coulter). cDNA was then synthesized using 5' RACE (rapid amplification of cDNA ends), adding distinct 3' and 5' template switch oligo adapters to total cDNA. First, the TSoligo2_polydT primer was

incubated with mRNA to append the 3' oligonucleotide. Next, first-strand cDNA synthesis and template switching to A-tag TSO was performed using SMARTseq reagents. cDNA was subsequently amplified with TSO_FWD and TS_Oligo_2_REV primers. Excess oligos and dNTPs were removed from amplified cDNA with EXO-CIP cleanup kit (New England BioLabs). All primer sequences are listed in Supplementary Table 5.

**Immunoglobulin enrichment and sequencing.** Heavy and light chain variable regions were enriched by amplifying cDNA with TSO_FWD and IgA/IgG_REV or IgK/IgL_REV primer pools. An aliquot of enriched product was used to prepare Nextera libraries with Unique Dual Indices (Illumina) and sequenced using $2 \times 150$ paired-end reads on an Illumina MiSeq. Separate aliquots were used for IG production; RATP-Ig is a modular system and can produce single combined or separate HC/LC cassettes. All primer sequences are listed in Supplementary Table 5.

**Cassette fragment synthesis.** Final linear cassettes include CMV, and HC/LC-TBGH polyA fragments. To isolate cassette fragments and introduce 15–20 base-pair overlaps, amplicons were first synthesized by PCR (CMV_FWD + CMV_TSO_REV, LC_FWD + TBGH_REV, HC_FWD + TBGH_polyA_REV). Crude PCR products were run on a 1% agarose gel and fragments of the correct size were extracted using the Thermo gel extraction and PCR cleanup kit (ThermoFisher Scientific). Gel-extracted products were digested with DpnI (New England Biolabs) to further remove any possible contaminating plasmid. These fragment templates were then further amplified with the same primers in large batches to create final working stocks of each cassette fragment. All primer sequences are listed in Supplementary Table 5.

**Cassette assembly.** Each cassette component (CMV, HC/LC TBGH, IRES, and IG enrichment product) contains overlapping 5' and/or 3' ends that allow for overlapping-assembly into a single linear strand of DNA. Overlapping sequences facilitate precision ligation prior to the final whole-cassette amplification to produce large quantities of DNA for transfection in 96-well microtiter plates.

Single Cassette: Enriched variable regions were assembled into linear expression cassettes in two sequential ligation reactions. The first reaction assembles CMV-TSO, TSO-V-LC, and KC-IRES fragments into part 1 and IRES-TSO, TSO-V-HC, and IgGC-TBGH fragments into part 2 using NEBuilder HIFI DNA Assembly Mastermix (New England BioLabs). Following reaction 1, parts 1 and 2 were combined into a single reaction 2 and ligated into a single cassette.

Separate cassettes: Enriched variable regions were assembled into linear expression cassettes by ligating CMV-TSO, TSO-V-C, and C-TBGH fragments using NEBuilder HIFI DNA Assembly Mastermix (New England BioLabs). Assembled cassettes were amplified using CMV_FWD and TBGH_REV primers. Amplified linear DNA cassettes encoding monoclonal heavy and light chain genes were co-transfected into Expi293 cells in 96-well deep-well plates using the Expi293 Expression System (ThermoFisher Scientific, catalog #: A41249) according to the manufacturer's protocol. Microtiter cultures were incubated at 37 degrees and 8% $CO_2$ with shaking at 1100 RPM for 5–7 days before supernatants were clarified by centrifugation and harvested. It is important to note that supernatant IgG titers were not calculated but were only verified to reach a minimum cutoff value for functional assays, limiting our ability to compare potency between antibodies.

## Droplet-based single cell isolation and sequencing

Antigen-specific memory B cells were sorted as described above. Cells from two separate sorts were pooled in a single suspension and loaded on the 10x Genomics Chromium instrument with reagents from the Next GEM Single Cell 5' Kit v1.1 following the manufacturer's protocol

to generate total cDNA. Heavy and light chains were amplified from the cDNA using IgG_REV, IgA_REV, IgK_REV or IgL_REV primers (Supplementary Table 5) with the addition of Illumina sequences[46]. The Illumina-ready libraries were sequenced using 2 × 300 paired-end reads on an Illumina MiSeq. Hashing oligonucleotides were amplified and sequenced from the total cDNA according to the 10x Genomics protocol.

## V(D)J sequence analysis

For cells processed via RATP-Ig, reads were demultiplexed using a custom script and candidate V(D)J sequences were generated using BALDR[47] and filtered for quality using a custom script (https://github.com/scharch/filterBALDR, doi: 10.5281/zenodo.7349126). Briefly, this script removes incomplete contigs (based on length and/or absence of a detected J gene) and those with very low read support. The sequences passing the filter were annotated using SONAR v4.2[20] in single-cell mode, using matching V genes and 80% nucleotide identity in CDR3 for both heavy and light chains to define clonality. IMGT gene nomenclature[48] was used to identify V, D, and J genes.

For cells processed via the 10x Genomics Chromium device, reads from the hashing libraries were processed using cellranger (10x Genomics). The resulting count matrix was imported into Seurat[49] and the sample of origin called using the HTODemux function. Paired-end reads from V(D)J libraries were preprocessed, merged and annotated using the default settings in SONAR (single-cell mode, with UMI detection and processing).

For all datasets, nonproductive rearrangements were discarded, as were any cells with more than one productive heavy or light chain. Cells with an unpaired heavy or light chain were included in calculations of SHM and gene usage statistics but were excluded from assessments of clonality and determination of public clones. Public clones were determined by using the cluster_fast command in usearch[50] to cluster CDR H3 amino acid sequences at 80% identity. Where relevant, all clonally related B cells in a single individual were included in a public clone, even if not all were directly clustered together in the vsearch analysis. While light chain V genes and CDR3 were not used to define public clones, they are reported when we found a consistent signature within a public clone. Logograms for public clones CDRH3s were generated using the `ggseqlogo` package in R (https://omarwagih.github.io/ggseqlogo/ commit `4adc8f2`).

Control repertoires were calculated using data from three individuals sequenced for the Human Immunome Project[26]. Neither the control individuals nor those in our study were genotyped, and data on ethnicity was not collected. In addition, the control repertoires were sequenced in bulk, using a variety of PCR and sequencing strategies that differ from those used here. Nonetheless, this represents the best available benchmarking, and similar control strategies are common[9,13].

## Statistical methods

To test for differences between different variants in antibody binding titers from serum we used Friedman's test followed by Dunn's test with multiple-comparison correction (MSD-ECLIA data) or Friedman's test followed by multiple comparisons with a two-stage linear step-up procedure of Benjamini, Krieger and Yekutieli (for cell surface S-binding data) using Graphpad Prism version 9.4.0.

To test for differences in epitope competition levels, we used an unpaired, two-tailed t-test (Graphpad Prism version 8.3.1).

To test for difference in V gene usage between groups, we conducted a Kruskal–Wallis test using 'kruskal.test' in R version 4.0.3. This was followed by a Dunn test with multiple testing correction using 'dunn.test' from the package dunn.test (version 1.3.5, available from cran.r-project.org).

To test for differences in somatic hypermutation levels, we used an unpaired, two-tailed Mann-Whitney U test (Graphpad Prism version 9.4.1).

## Reporting summary

Further information on research design is available in the Nature Portfolio Reporting Summary linked to this article.

## Data availability

The raw sequencing data have been deposited in the SRA under Bio-Project PRJNA832903.

Processed and annotated V(D)J sequences are included as a Supplementary Dataset. Source data are provided with this paper.

## Code availability

Custom code for this analysis is available from GitHub at https://github.com/scharch/filterBALDR (https://doi.org/10.5281/zenodo.7349126).

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

## Acknowledgements

The authors would like to thank the members of the VRC 200 Study Team for their role in collecting samples that were used in this study: Lesia Drupolic, Lasonji Holman, Maria Burgos Florez, Charla Andrews, Britta Flach, Emily Coates, Obrimpong Amoa-Awua, Jennifer Cunning-ham, Pamela Costner, Floreliz Mendoza, William Whalen, Jamie Saun-ders, Laura Novik, Aba Eshun, Anita Arthur, Xiaolin Wang, Karen Parker, Abidemi Ola, Catina Evans, Jennifer Phipps, Pernell Williams, Justine Jones, Jackie Stephens, Jumoke Gbadebo, Preeti Apete, Renunda Hicks, LaShawn Requillman, Alison Beck, Seemal Awan, Richard Wu, Priya Kamath, Olga Trofymenko, Sarah Plummer, Nina Berkowitz, Olga Vasi-lenko, and Iris Pittman. The authors also thank Dr. Steven De Rosa (Fred Hutchinson Cancer Center) for providing a 28-color flow cytometry panel which we modified for our study and David Ambrozak for assis-tance with cell sorting. The authors thank Rodrigo Matus Nicodemus for his assistance in developing primers for RATP-Ig. Funding: This work was funded in part by the Intramural Research Program of the Vaccine Research Center, National Institute of Allergy and Infection Disease, National Institutes of Health.

## Author contributions

Conceptualization: N.S.L., C.A.S., D.C.D. Data curation: M.M., C.A.S. Formal Analysis: N.S.L., M.M., T.S.J., D.A.W., L.W., K.B., S.R.N., S.O.C., L.S., K.L.B., C.A.S. Investigation: N.S.L., M.M., T.S.J., D.A.W., A.R.H., L.W., K.B., W.P.B., S.D.S., D.M., C.G.L., B.Z., K.L.B., J.R.T., R.L.D., L.P., L.S., J.W., C.A.T., GS Methodology: T.S.J., D.C.D. Resources: E.S.Y., Y.Z., S.O.D., M.C., A.S., K.L., W.S., R,K., A.B., T.Z., J.R., S.V., A.A., L.N., A.W., I.G., M.G., I.T.T., E.P., T.J.R. Supervision: A.P., J.M., N.A.D.R., M.G., R.A.K., P.D.K., A.B.M., S.A., T.W.S., I.L., J.R.M., N.J.S., C.A.S., D.C.D. Visualization: N.S.L., M.M., T.S.J., D.A.W., K.L.B., C.A.S., D.C.D. Writing – original draft: N.S.L., M.M., T.S.J., C.A.S., D.C.D. Writing – review & editing: all authors

## Funding

## Competing interests

The authors declare no competing interests.

## Additional information

[1]Vaccine Research Center, National Institute of Allergy and Infectious Diseases, National Institutes of Health, Bethesda, MD 20892, USA. [2]Infectious Disease Unit, Sheba Medical Center, Ramat Gan 5262112, Israel. [3]Sackler Medical School, Tel Aviv University, Tel Aviv 6997801, Israel. [4]Department of Medicine, University of Minnesota Medical School, Minneapolis, MN 55455, USA. [5]Minnesota Department of Health, St Paul, MN 55164, USA. [6]Clinical Microbiology, Sheba Medical Center, Ramat-Gan 5262112, Israel. [7]These authors contributed equally: Noemia S. Lima, Maryam Musayev, Timothy S. Johnston, Danielle A. Wagner. ✉e-mail: chaim.schramm@nih.gov; ddouek@nih.gov

