## [Peer Review File · Nature Communications]

Primary exposure to SARS-CoV-2 variants elicits convergent epitope specificities, immunoglobulin V gene usage and public B cell clonesREVIEWER COMMENTS

Reviewer #1 (Remarks to the Author):

Summary

In this study, variant specific B cells from convalescent patient samples following WA-1, Beta and Gamma SARS-CoV-2 infections are sequenced and the antibodies characterized for binding, cross-reactivities and neutralization. The authors introduce their RAPT-Ig pipeline which allows cloning-free, high-throughput recovery of linear transfection cassettes for antibody production from single-cell sorted B cells. They find cross reactive B cells across infecting variants and more multationally distant Delta and Omicron variants and characterize conserved epitope immunodominance. Integrating their RAPT-Ig data with paired 10X single cell sequencing antibody repertoires, they identify public antigen specific B cell clones and convergent V gene usage across variant infections.

Major Comments

1. The study integrates paired heavy and light chain sequences of antibodies elicited following SARS-CoV-2 variant infections and applies several methods to obtain antigen specific B cells and different sorts. To the reader it is at times confusing to have an overview about which samples are compared in an analysis. Therefore the manuscript might profit from a more detailed graphical overview figure, similar to Extended Figure 4A depicting the downstream analysis pipeline, highlighting the antigen probes used for each sample.
2. MSD-ECLIA was applied to measure serum binding to cell surface bound Spike proteins (S) as well as soluble S trimers and RBDs of each variant (Extended Figure 2A, Figure 1A). Significant p-values are reported for binding to Omicron BA-1S and Beta RBD for WA-1 immunized samples. From visual inspection also other variants have similarly decreased binding efficiency. Could the authors provide p values for these comparisons or are they all insignificant (line 77) ? Additionally the trends observed between the assays for cell surface S and S-2P seem inconsistent. Could the authors discuss in more detail the necessity for a cell surface S display and the influence on the results? Furthermore it would be interesting to see the S-2P and RBD binding of the sera against Omicron and Delta variants as in Figure 1A.
3. Figure 1C: Addition of Gamma samples against homologous spike protein would be necessary to make the claim that there seems to be uniform epitope targeting across variants. In Figure S2C and D, why are sites H to K not shown for Gamma? If the Antibodies tested do not bind to Gamma RBD this should be mentioned similarly as in Figure 1C.
4. Restimulation potential of memory CD4+ and CD8+ T cells was characterized and showed variant independent epitope usage with strong activation for whole spike libraries and weak

restimulation when only peptide fragments containing variant epitopes were used. When describing this assay, the authors should explain more clearly in the main text what peptide pools were used and how they were constructed (Figure 1D). How do the authors explain the low restimulation of CD8+ T cells from Wuhan convalescent samples compared to the other variants?

5. RAPT-IG is a good method for cloning free assembly of IgG sequences. The cloning-free approach to directly obtain antibodies from cell culture supernatant without plasmid assembly is desirable. However, the manuscript focuses on its high-throughput, which is not really comparable to the high-throughput obtained with other methods such as 10X single cell sequencing. A more in depth description of the workflow and emphasis on the fast and direct generation of transfection-ready linear cassettes. The authors should mention how efficient this method is (i.e. how many cells were initially sorted to give rise to the 509 single cells recovered for the analysis?)
6. When sorting antigen specific B cells for RAPT-IG, the authors used WA-1 and Beta S-2P/NTD and RBD probes. What is the reason WA-1 and Beta have been chosen to select cross specific antibodies and Gamma not included in the sort? Only few beta-Wuhan cross reactive antibodies from gamma specific samples could be found and only AB14.1 seems to be of interest, while this sample showed high cross reactivity and cross neutralization in Figure 1A and B. Are the differences observed due to the lack of gamma specific probes or the low coverage by RAPT-Ig?
7. When comparing clonally expanded cells, could common repertoire features be observed compared to singletons? Is there a relationship between clonal expansion and neutralization? Integrating clonal expansion in Figure 2C or highlighting neutralizing clones in the donut plots of Figure 2D would be very interesting since clonal expansion has been linked to antigen specificity and neutralization.
8. Having obtained many RBD neutralizing antibodies, a competition assay as performed for the serum would be really valuable to relate epitope bins of individual neutralizing antibodies to serum bins. Which RBD sites are targeted by neutralizing antibodies?
9. The authors use 10X single cell sequencing to obtain paired heavy and light chain information of antigen specific B cells from all samples. The recovered numbers of paired heavy and light chains seem very low compared to other datasets obtained by this pipeline. Could the authors discuss reasons for the low recovery of cells? In the main text, the authors should describe which clonotyping strategy was applied. According to the methods, clonotypes were clustered at 80% CDRH3 homology. From the public clones reported in Figure 5A, the same VH gene as well as CDRH3 length seem to be defining characteristics, is this a side effect of the 80% CDRH3 similarity clustering? Was the same clonotyping strategy applied for RAPT-Ig clonotyping in Figure 2D?
10. More analysis of the obtained single cell sequencing dataset would be of interest. For example, similar expansion donut plots as in Figure 2A.
11. Why were no Gamma specific probes used for the Beta convalescent samples and no Beta probes for the gamma convalescent samples? (Figure 3A)
12. For most comparisons there is only significant enrichment of a V gene compared to one variant infection (Figures 3C, Extended Figure 7 and 8). Can the authors elaborate how they define a convergent response across variants (line 170)? It seems from this data that the V gene usage compared to control seems pretty variant specific.
13. The authors compared the frequency of V genes associated with Beta or Gamma binding and found no differences in the variant specific binding repertoires (Figure 3D&E). It should be mentioned which specific V genes these are. No differences could be found for WA-1 and Beta binders within the Beta infected samples which the authors claim is due to the

fact that these genes are mainly enriched in neutralizing antibodies. In this study, neutralizing antibodies have been found. Are these enriched genes represented in those (line 180)?

14. The data generated in this study is very valuable and the characterisation of antigen specificities and cross reactivities of high interest for many studies. From this manuscript, the antibody sequences recovered are not accessible. Do the authors intend to add a more complete table with antibody sequences, annotated V genes and specificities for all characterized antibodies generated by RAPT-IG and 10X sequencing or add this information to a publicly available database? This would not only allow reproduction of the presented results but also be highly valuable for integration into other studies investigating cross reactivities and naturalization of SARS-CoV-2 variants.

Minor Comments

- Figure 1C: Do figure legends indicate whether WA-1 and Beta refer to proteins or infection cohorts? This should be more clearly labeled across figures since the same colors are used interchangeably.
- Line 81: Reference to Figure 2A instead of 1A.
- Figure 3: Addition of patient symbols to dots would allow for comparison of antigen specific cells to serum titers shown in Fig 1 and visualize correlation between titers and antigen specific cell fraction.
- Supplementary Table 1-3: Neutralization should be reported for all antibodies if measured. Otherwise it's not clear why the selected ab in figure 2C are shown (Line 122). Is neutralization shown in figure 2C at 4 or 6 fold dilution? Data of this assay should be available.
- The authors should briefly give background information of the pre pandemic control samples used in Figure 3. How comparable is the genetic background of these samples to the convalescent patient samples in this study?
- Line 450: Methods for RAPT-IG B cell sorting should refer to Extended Figure 4B and C instead of only B. Also the 10X repertoire sequencing sorting strategy should refer to Extended Figure 4B and D.

Reviewer #2 (Remarks to the Author):

Lima and colleagues utilize a novel approach for Ig chain sequencing and mAb generation from single Ag specific B cells to characterise the antibody response to infection with different SARS-CoV-2 variants. Responses from individuals with primary infection with beta, gamma or the ancestral strain are characterised in the period between ~3-5 weeks post symptom onset in a small cohorts recruited in 2020 (ancestral) or early 2021 (beta/gamma). mAbs are also assessed for cross-reactivity to the more recent delta and omicron variants. Convergent/stereotypic responses among the individuals are found for all strains. They also explore the epitope directed response to the different variants by stimulation of T cells with different peptide pools and for the B cells via SPR and they found that the epitope immunodominance hierarchy is unchanged across the different variants.

The strengths of the study lie in the extent of the pairing of Ig sequencing and functional characterisation of the Ag-specific response, via the mAbs, to the three different variants. The study may be limited by cohort sizes for each variant and the overall number of single cells analysed, however if the convergent/stereotyped responses can be detected in small cohorts then this suggests that they may only be more frequent if the cohort sizes were larger.

The manuscript contributes a unique perspective on the impact of SARS-CoV-2 variants on the 'public' clones that were noted in early studies of predominantly the ancestral strain. Their approach to higher throughput mAbs from single cell sorting is also likely to be of interest to the field. The manuscript is well written, and methodologies and analysis appear robust and the conclusions are well supported. Methods for the most part include sufficient detail or cite references with the appropriate details, a little more detail on the workflow of the custom scripts for the repertoire sequencing could be provided in absence of providing a link to the code. Some minor improvements could be made to data presentation and analysis and authors should consider deposit/release of sequence data for the ~1k Ag-specific single cells to support the manuscript.

Specific comments

- line 66 – define PBMC
- line 68 – can you confirm that they were naïve to both vaccine and infection and how was this determined?
- Line 187 – for the SHM differences between the single probe (10x) and the double-probe (RAPT-Ig) did you consider whether there were differences from the need to generate contigs/consensus via BALDR for RAPT-Ig versus the more 'direct' merging of the mate pair reads from the 10x? Were there any biases that may have been introduced by the quality filtering with the custom script compared to the 10x derived data which doesn't appear to have undergone any pre-processing prior to analysis with SONAR?
- Line 201 – 'which data is available' -> 'which data are available'?
- Line 215 – with the repeated usage of IGHV3-30 (also relevant to IGHV1-69) among the public clones, did you undertake any analysis considering the if allelic variants are important or was this not possible without deeper sequencing of the individual repertoires to undertake genotype/haplotype inference?
- Line 504 – details of the quality filtering implemented by the custom script for the contigs from BALDR
- Line 509 – any pre-processing?
- Line 966 – deposit of sequencing data?
- Figure 1 – consider using the per donor shapes for data points from sub-panel A for panels C and D?
- Figure 2 – the continuous scale for sub-panel D makes it difficult to work out what the counts are. Also, the tick marks in the scale appear not to be aligned to the labels. A discrete colour scale may be more easily interpreted.
- Figure 3 – sub-panel C – it appears that zeros aren't being plotted when a gene is absent from a subset? What is the statistical test, couldn't see it in the legend, how was the absence of a gene in a sub-group treated for the stats?
- Figure 4 – in sub-panel A it is hard to see the mean +/- std dev shown with the error bars

among all the points. Maybe try a different colour for the error bars or shift them to the side of the points to make more visible.

(Reviewer #1)

Summary

In this study, variant specific B cells from convalescent patient samples following WA-1, Beta and Gamma SARS-CoV-2 infections are sequenced and the antibodies characterized for binding, cross-reactivities and neutralization. The authors introduce their RAPT-Ig pipeline which allows cloning-free, high-throughput recovery of linear transfection cassettes for antibody production from single-cell sorted B cells. They find cross reactive B cells across infecting variants and more multationally distant Delta and Omicron variants and characterize conserved epitope immunodominance. Integrating their RAPT-Ig data with paired 10X single cell sequencing antibody repertoires, they identify public antigen specific B cell clones and convergent V gene usage across variant infections.

Major Comments

1. The study integrates paired heavy and light chain sequences of antibodies elicited following SARS-CoV-2 variant infections and applies several methods to obtain antigen specific B cells and different sorts. To the reader it is at times confusing to have an overview about which samples are compared in an analysis. Therefore the manuscript might profit from a more detailed graphical overview figure, similar to Extended Figure 4A depicting the downstream analysis pipeline, highlighting the antigen probes used for each sample.

We thank the reviewer for pointing out this problem and have substantially re-designed Extended Figure 4A to make it clear which samples had which infection history, their sorting strategy, and subsequent sequencing and Ig expression approach.

2. MSD-ECLIA was applied to measure serum binding to cell surface bound Spike proteins (S) as well as soluble S trimers and RBDs of each variant (Extended Figure 2A, Figure 1A). Significant p-values are reported for binding to Omicron BA-1S and Beta RBD for WA-1 immunized samples. From visual inspection also other variants have similarly decreased binding efficiency. Could the authors provide p values for these comparisons or are they all insignificant (line 81-82)?

The p values for these comparisons are insignificant and we have added clarification of this point to the text.

Additionally the trends observed between the assays for cell surface S and S-2P seem inconsistent. Could the authors discuss in more detail the necessity for a cell surface S display and the influence on the results?

Thank you for the opportunity to clarify these points. The cell surface Spike binding data (shown in Figure 1A) were measured by flow cytometry, while the S-2P binding data (shown in Extended Figure 2A) were acquired by MSD-ECLIA. The inconsistent trend between both methods is seen exclusively with variants that exhibit low serum binding titers and we believe this may be due to the lower sensitivity of the flow cytometry-based method compared to the MSD method. However, the cell surface Spike binding assay correlates better with neutralization, probably because both assays use the same Spike construct, while the MSD assay uses the S-2P stabilized version of the Spike. There may be differences in the epitopes exposed between WA1 Spike and S-2P Spike due to the 2P stabilization. For these reasons, we believe that showing the data for both assays add value to interpreting our findings and have explained this in the text (lines 75-79).

Furthermore it would be interesting to see the S-2P and RBD binding of the sera against Omicron and Delta variants as in Figure 1A.

We agree and have now done these experiments and added the data to the Extended Figure 2A.

3. Figure 1C: Addition of Gamma samples against homologous spike protein would be necessary to make the claim that there seems to be uniform epitope targeting across variants.

We have now done these experiments and included the data. Our new data support our manuscript's conclusion that Gamma-infected donors show comparable patterns of reactivity to homologous Spike as WA1- or Beta-infected donors.

In Figure S2C and D, why are sites H to K not shown for Gamma? If the Antibodies tested do not bind to Gamma RBD this should be mentioned similarly as in Figure 1C.

The reviewer raises an important point. These epitopes are disrupted on the Delta Spike protein. We have made clarifications to the figure and added the suggested text to the figure legend.

4. Restimulation potential of memory CD4+ and CD8+ T cells was characterized and showed variant independent epitope usage with strong activation for whole spike libraries and weak restimulation when only peptide fragments containing variant epitopes were used. When describing this assay, the authors should explain more clearly in the main text what peptide pools were used and how they were constructed (Figure 1D).

These pools are described in the Methods and Supplementary Tables; we have now added explicit pointers to the Supplementary Tables in the paragraph describing these results (lines 104 and 109), such that they may be more easily identified by the reader.

How do the authors explain the low restimulation of CD8+ T cells from Wuhan convalescent samples compared to the other variants?

The difference noted by the reviewer is driven primarily by one Beta-infected outlier; the difference between Gamma- and WA1-infected donors rare similar to the differences seen in their IG responses. We have added a sentence to the text (lines 107-108) clarifying this.

5. RAPT-IG is a good method for cloning free assembly of IgG sequences. The cloning-free approach to directly obtain antibodies from cell culture supernatant without plasmid assembly is desirable. However, the manuscript focuses on its high-throughput, which is not really comparable to the high-throughput obtained with other methods such as 10X single cell sequencing.

Although 10x sequencing is much higher throughput with respect to obtaining IG sequences, those sequences must then be chemically synthesized, transfected and expressed before functional data can be obtained. Hence the name of our method, Rapid Assembly, Transfection, and Production of Immunoglobulins, which reflects these differences compared to others such as 10x.

A more in depth description of the workflow and emphasis on the fast and direct generation of transfection-ready linear cassettes.

This is a great suggestion and we have added more detail to the methods.

The authors should mention how efficient this method is (i.e. how many cells were initially sorted to give rise to the 509 single cells recovered for the analysis?)

The number 509 represents all such sorted cells that were present among the three PBMC samples; the efficiency of the method should be evaluated with that as the starting point, as stated in lines 121 and 124.

6. When sorting antigen specific B cells for RAPT-IG, the authors used WA-1 and Beta S2P/NTD and RBD probes. What is the reason WA-1 and Beta have been chosen to select cross specific antibodies and Gamma not included in the sort?

We limited the probes to two variants in order to avoid competition effects (which we consistently observe) that would have made it more difficult to sort the antigen-specific cells. As two of the three donors selected for RAPT-Ig were Beta-infected, we elected to use WA1 and Beta probes for all three for consistency.

Only few beta-Wuhan crossreactive antibodies from gamma specific samples could be found and only AB14.1 seems to be of interest, while this sample showed high cross reactivity and cross neutralization in Figure 1A and B. Are the differences observed due to the lack of gamma specific probes or the low coverage by RAPT-Ig?

This really is an excellent point and one about which we can only speculate. We have added a sentence noting this possibility (lines 134-136).

7. When comparing clonally expanded cells, could common repertoire features be observed compared to singletons?

We thank the reviewer for this suggestion. Upon analysis of expanded vs unexpanded clones, we discovered that the former are predominantly derived from two V_H genes (treating closely related IGVH3-30 subfamily genes as indistinguishable). We have added this in lines 154-160.

Is there a relationship between clonal expansion and neutralization? Integrating clonal expansion in Figure 2C or highlighting neutralizing clones in the donut plots of Figure 2D would be very interesting since clonal expansion has been linked to antigen specificity and neutralization.

Only 2 neutralizers are from expanded clones; the other 12 are singletons. We have noted this fact in line 158.

8. Having obtained many RBD neutralizing antibodies, a competition assay as performed for the serum would be really valuable to relate epitope bins of individual neutralizing antibodies to serum bins. Which RBD sites are targeted by neutralizing antibodies?

This is a great question. However, the SPR technique used for the competition assay performed in serum requires high amounts of antibodies, which are easily achievable in serum and by monoclonal antibody expression from plasmids. Our RAPT-Ig method is a cloning-free technique that allows expression of antibodies in low scale for rapid, high-throughput screening and characterization. A more in-depth characterization of the isolated mAbs, including epitope mapping, would require to obtain plasmids for antibody expression. This approach would delay the publication of the manuscript and we believe that it is not crucial for the message that want to convey.

9. The authors use 10X single cell sequencing to obtain paired heavy and light chain information of antigen specific B cells from all samples. The recovered numbers of paired heavy and light chains seem very low compared to other datasets obtained by this pipeline. Could the authors discuss reasons for the low recovery of cells?

The total number of cells recovered is low primarily because antigen-specific cells are relatively rare – the frequency at which we observe these cells (0-1.5% of IgG⁺ B cells) is consistent with other literature. We have added the total number of cells sorted on line 183 to make this clear. We note that the yield of recovered sequences relative to input cells was, indeed, unexpectedly low for the Beta-infected group; however, there are no remaining samples with which to repeat those experiments.

In the main text, the authors should describe which clonotyping strategy was applied.

This has been added, lines 152-154.

According to the methods, clonotypes were clustered at 80% CDRH3 homology. From the public clones reported in Figure 5A, the same V_H gene as well as CDRH3 length seem to be defining characteristics, is this a side effect of the 80% CDRH3 similarity clustering? Was the same clonotyping strategy applied for RAPT-Ig clonotyping in Figure 2D?

We apologize for the confusion and have clarified the differences in how biological vs public clones were calculated in lines 230-234.

10. More analysis of the obtained single cell sequencing dataset would be of interest. For example, similar expansion donut plots as in Figure 2A.

We have now added the clonality donut plots for all the samples subjected to single cell sequencing by 10x Genomics to Extended Data Figure 7.

11. Why were no Gamma specific probes used for the Beta convalescent samples and no Beta probes for the gamma convalescent samples? (Figure 3A)

We did not have enough PBMC to conduct multiple sorts of these samples and, as noted above, including probes from more than two variants in a single sort would have resulted in too much competition between probes. Therefore, we chose to limit the assay to the homologous variant probe and WA1 for these groups.

12. For most comparisons there is only significant enrichment of a V gene compared to one variant infection (Figures 3C, Extended Figure 7 and 8). Can the authors elaborate how they define a convergent response across variants (line 170)? It seems from this data that the V gene usage compared to control seems pretty variant specific.

We agree that our description of these results was not clear. In addition, there was also a methodological error, pointed out by the other reviewer, which has now been corrected. Figure 3C and ED Figs 8-9 (previously ED 7-8) have been revised, as has the accompanying text on lines 191-197.

13. The authors compared the frequency of V genes associated with Beta or Gamma binding and found no differences in the variant specific binding repertoires (Figure 3D&E). It should be mentioned which specific V genes these are.

We apologize for the oversight; the specific V genes are now noted on lines 201-202.

No differences could be found for WA-1 and Beta binders within the Beta infected samples which the authors claim is due to the fact that these genes are mainly enriched in neutralizing antibodies. In this study, neutralizing antibodies have been found. Are these enriched genes represented in those (line 180)?

They are not, likely because the neutralizing antibodies we isolated were specifically derived from the cross-binding subset that was sorted for RAPT-Ig and thus will exclude antibodies that rely on Y501. We have added text in lines 203-206 clarifying this.

14. The data generated in this study is very valuable and the characterisation of antigen specificities and crossreactivities of high interest for many studies. From this manuscript, the antibody sequences recovered are not accessible. Do the authors intend to add a more complete table with antibody sequences, annotated V genes and specificities for all characterized antibodies generated by RAPT-IG and 10X sequencing or add this information to a publicly available database? This would not only allow reproduction of the presented results but also be highly valuable for integration into other studies investigating cross reactivities and naturalization of SARS-CoV-2 variants.

We apologize for the oversight, sequences are now included as Supplementary Table 6. Specificities (for RAPT-Ig antibodies only, as we did not synthesize any sequences from the 10x data) are already included in Supplementary Tables 3-5 (previously Supplementary Tables 1-3).

Minor Comments

- Figure 1C: Do figure legends indicate whether WA-1 and Beta refer to proteins or infection cohorts? This should be more clearly labeled across figures since the same colors are used interchangeably.

Thanks, we have done so.

- Line 81: Reference to Figure 2A instead of 1A.

Done.

- Figure 3: Addition of patient symbols to dots would allow for comparison of antigen specific cells to serum titers shown in Fig 1 and visualize correlation between titers and antigen specific cell fraction.

Thanks for the suggestion. We have added the symbol shapes to each donor on panels C and D of Figure 3A and also to Figure 1 and Extended Figure 2C-D.

- Supplementary Table 1-3: Neutralization should be reported for all antibodies if measured. Otherwise it's not clear why the selected ab in figure 2C are shown (Line 122). Is neutralization shown in figure 2C at 4 or 6 fold dilution? Data of this assay should be available.

Figure 2C presents all the antibodies that showed neutralization. All other antibodies did not neutralize.

- The authors should briefly give background information of the pre pandemic control samples used in Figure 3. How comparable is the genetic background of these samples to the convalescent patient samples in this study?

This is an important point and is now addressed in the Methods (lines 580-585).

- Line 450: Methods for RAPT-IG B cell sorting should refer to Extended Figure 4B and C instead of only B. Also the 10X repertoire sequencing sorting strategy should refer to Extended Figure 4B and D.

We have fixed these errors, thanks.

(Reviewer #2)

Lima and colleagues utilize a novel approach for Ig chain sequencing and mAb generation from single Ag specific B cells to characterise the antibody response to infection with different SARS-CoV-2 variants. Responses from individuals with primary infection with beta, gamma or the ancestral strain are characterised in the period between ~3-5 weeks post symptom onset in a small cohorts recruited in 2020 (ancestral) or early 2021 (beta/gamma). mAbs are also assessed for cross-reactivity to the more recent delta and omicron variants. Convergent/stereotypic responses among the individuals are found for all strains. They also explore the epitope directed response to the different variants by stimulation of T cells with different peptide pools and for the B cells via SPR and they found that the epitope immunodominance hierarchy is unchanged across the different variants.

The strengths of the study lie in the extent of the pairing of Ig sequencing and functional characterisation of the Ag-specific response, via the mAbs, to the three different variants. The study may be limited by cohort sizes for each variant and the overall number of single cells analysed, however if the convergent/stereotyped responses can be detected in small cohorts then this suggests that they may only be more frequent if the cohort sizes were larger.

The manuscript contributes a unique perspective on the impact of SARS-CoV-2 variants on the 'public' clones that were noted in early studies of predominantly the ancestral strain. Their approach to higher throughput mAbs from single cell sorting is also likely to be of interest to the field. The manuscript is well written, and methodologies and analysis appear robust and the conclusions are well supported. Methods for the most part include sufficient detail or cite references with the appropriate details, a little more detail on the workflow of the custom scripts for the repertoire sequencing could be provided in absence of providing a link to the code. Some minor improvements could be made to data presentation and analysis and authors should consider deposit/release of sequence data for the ~1k Ag-specific single cells to support the manuscript.

We thank the reviewer for the thoughtful summary of our work.

Specific comments

- line 66 – define PBMC

Done.

- line 68 – can you confirm that they were naïve to both vaccine and infection and how was this determined?

We have clarified in the text that we meant “no known prior exposures” (lines 68-69). In addition, samples were collected before vaccines were widely available.

- Line 187 – for the SHM differences between the single probe (10x) and the double-probe (RAPT-Ig) did you consider whether there were differences from the need to generate contigs/consensus via BALDR for RAPT-Ig versus the more ‘direct’ merging of the mate pair reads from the 10x? Were there any biases that may have been introduced by the quality filtering with the custom script compared to the 10x derived data which doesn’t appear to have undergone any pre-processing prior to analysis with SONAR?

This is an interesting and insightful question. While we have not investigated it systematically, we have on a number of occasions sequenced both the IG-enriched cDNA and the final cassettes that are transfected for expression; in all cases the sequences have matched identically. In general, the filtering of the BALDR output is very light, and primarily removes contigs with very low read support (see below for more about the script). We are thus reasonably confident that sequencing and analysis strategy is not introducing bias into the final data.

- Line 201 – ‘which data is available’ -> ‘which data are available’?

Fixed, thanks.

- Line 215 – with the repeated usage of IGHV3-30 (also relevant to IGHV1-69) among the public clones, did you undertake any analysis considering the if allelic variants are important or was this not possible without deeper sequencing of the individual repertoires to undertake genotype/haplotype inference?

Unfortunately, no, this kind of analysis is not possible with the data that we have. We have added a note of this limitation in lines 330-332.

- Line 504 – details of the quality filtering implemented by the custom script for the contigs from BALDR

This script is now included in the Software Reporting form submitted with this revision and has been made available at <https://github.com/scharch/filterBALDR>. Details have also been added to the methods, lines 561-562.

- Line 509 – any pre-processing?

The preprocessing was also done in SONAR; this has now been clarified.

- Line 966 – deposit of sequencing data?

Raw sequencing data has been deposited in the SRA and the review link submitted to the Editor.

- Figure 1 – consider using the per donor shapes for data points from sub-panel A for panels C and D?

Thanks for the suggestion. We have added the symbol shapes to each donor on panels C and D of Figure 1 and also to Extended Figure 2C-D and Figure 3A.

- Figure 2 – the continuous scale for sub-panel D makes it difficult to work out what the counts are. Also, the tick marks in the scale appear not to be aligned to the labels. A discrete colour scale may be more easily interpreted.

Figure 2D was edited as the reviewer suggestion.

- Figure 3 – sub-panel C – it appears that zeros aren’t being plotted when a gene is absent from a subset? What is the statistical test, couldn’t see it in the legend, how was the absence of a gene in a sub-group treated for the stats?

We thank the reviewer for catching this error, and the zeroes have now been properly added to the plots and considered in the statistical tests. Figure 3C and ED Figs 8-9 (previously ED 7-8) have been corrected, as has the text on lines 191-197.

- Figure 4 – in sub-panel A it is hard to see the mean +/- std dev shown with the error bars among all the points. Maybe try a different colour for the error bars or shift them to the side of the points to make more visible.

Thanks for the suggestion. Figure 4 was edited to show wider mean and std dev bars to improve visibility.

REVIEWERS' COMMENTS

Reviewer #1 (expertise in immunoglobulin repertoires):

The authors have done a great job of answering the extensive questions and comments. I am happy to support the manuscript for publication.

Reviewer #2 (expertise in immune receptor repertoire sequencing, virus-specific repertoires):

The revised submission addresses all points raised in my initial review.